# Photocatalytic Activity of the V_2_O_5_ Catalyst toward Selected Pharmaceuticals and Their Mixture: Influence of the Molecular Structure on the Efficiency of the Process

**DOI:** 10.3390/molecules28020655

**Published:** 2023-01-09

**Authors:** Sanja J. Armaković, Aleksandra Jovanoski Kostić, Andrijana Bilić, Maria M. Savanović, Nataša Tomić, Aleksandar Kremenović, Maja Šćepanović, Mirjana Grujić-Brojčin, Jovana Ćirković, Stevan Armaković

**Affiliations:** 1University of Novi Sad, Faculty of Sciences, Department of Chemistry, Biochemistry and Environmental Protection, 21000 Novi Sad, Serbia; 2Center for Solid State and New Materials, Institute of Physics Belgrade, 11000 Belgrade, Serbia; 3Laboratory of Crystallography, Faculty of Mining and Geology, University of Belgrade, Đušina 7, 11000 Belgrade, Serbia; 4Institute for Multidisciplinary Research, University of Belgrade, 11000 Belgrade, Serbia; 5University of Novi Sad, Faculty of Sciences, Department of Physics, 21000 Novi Sad, Serbia

**Keywords:** β-blocker, nadolol, pindolol, metoprolol, photocatalysis, nanomaterial characterization, DFT analysis

## Abstract

Due to the inability of conventional wastewater treatment procedures to remove organic pharmaceutical pollutants, active pharmaceutical components remain in wastewater and even reach tap water. In terms of pharmaceutical pollutants, the scientific community focuses on β-blockers due to their extensive (over)usage and moderately high solubility. In this study, the photocatalytic activity of V_2_O_5_ was investigated through the degradation of nadolol (NAD), pindolol (PIN), metoprolol (MET), and their mixture under ultraviolet (UV) irradiation in water. For the preparation of V_2_O_5_, facile hydrothermal synthesis was used. The structural, morphological, and surface properties and purity of synthesized V_2_O_5_ powder were investigated by scanning electron microscopy (SEM), X-ray, and Raman spectroscopy. SEM micrographs showed hexagonal-shaped platelets with well-defined morphology of materials with diameters in the range of 10–65 µm and thickness of around a few microns. X-ray diffraction identified only one crystalline phase in the sample. The Raman scattering measurements taken on the catalyst confirmed the result of XRPD. Degradation kinetics were monitored by ultra-fast liquid chromatography with diode array detection. The results showed that in individual solutions, photocatalytic degradation of MET and NAD was relatively insignificant (<10%). However, in the PIN case, the degradation was significant (64%). In the mixture, the photodegradation efficiency of MET and NAD slightly increased (15% and 13%). Conversely, it reduced the PIN to the still satisfactory value of 40%. Computational analysis based on molecular and periodic density functional theory calculations was used to complement our experimental findings. Calculations of the average local ionization energy indicate that the PIN is the most reactive of all three considered molecules in terms of removing an electron from it.

## 1. Introduction

The World Health Organization noted in its reports that pharmaceuticals are present in natural and treated water in concentrations from 0.1 to 0.05 mg/L [1]. Among all pharmaceuticals, a large increase was noted for β-blockers due to an increase in cardiovascular diseases [2]. Because of their moderately high solubility [3] and incomplete removal by wastewater treatment plants, β-blockers are persistent in water environments and they may show potential toxic properties against nontarget organisms, i.e., aquatic organisms and even human health. For example, metoprolol (MET, Figure 1) is one of the most used medicines in this class, and its usage increased four times in recent years [4]. Pindolol (PIN, Figure 1) possesses lipophilic properties and at natural pH is almost completely insoluble in water [5,6]. Nadolol (NAD, Figure 1) is another medicine from the β-blockers class, which is highly soluble in water and hydrolytically stable, and thanks to that property, is highly bioavailable in the environment [7,8]. Usually, β-blockers are present in the environment in combination with other β-blockers and various organic pollutants. Their mutual reactions may manifest additional health and environmental risks caused by unexpected interactions between them.

Conventional water treatment plants (WWTPs) lack the potential to remove emerging pollutants, such as pharmaceuticals, especially β-blockers [9]. The poor efficiency of conventional WWTPs in removing β-blockers is proven by the presence of β-blockers in various water organisms [10]. The average removal efficiency of these compounds is around 30–40% [11]. The degradation of individual β-blockers depends on various factors, such as temperature, pH value, compound’s polarity, cation-exchange properties, biodegradability, etc. [12]. Multiple strategies have been applied in their removals, such as adsorption [13], chemical treatment for tackling water contamination, and advanced oxidation processes (AOPs) [14,15]. However, to achieve easy operation, low equipment cost, and cheap raw materials, AOPs are highly desirable. In addition, AOP, such as photodegradation, is considered most commonly used due to its ability to remove most organic pollutants from water [16]. To prevent and eliminate their unexpected toxic properties, there is a need for new efficient methods and materials for β-blockers removal from the water environment. A slow degradation rate during photolysis requires catalysts activated by interaction with light, after which the degradation process is more efficient [8]. For more than ten years, semiconductor-based photocatalysts have focused on water purification because of their high potential for the degradation of pollutants [17]. Some latest photocatalysts used are nanocrystalline M-type hexaferrite Ca_0.5_Pb_0.5_-xYb_x_Zn_y_Fe_12-y_O_19_ synthesized by sol–gel autocombustion method [15], the C_3_N_4x_/AgO_y_@Co_1-x_Bi_1-y_O_7_ synthesized by the combination of sol–gel and annealing approaches [14], porous methacrylic organosilica materials (Ag@PMOS) synthesized by reducing the silver moieties on and in the surfaces of porous methacrylic organic silicates [18], V_2_O_5_ nanoparticles [19], V_2_O_5_ nanorods [20], etc.

Vanadium oxide is a compound with colossal potential in water treatment because of its capability of light absorption, lower bandgap (2.3 eV) than TiO_2_ and ZnO, chemical stability, and surface catalytic properties [21,22,23,24,25]. Following previously obtained results (Table 1), the V_2_O_5_ catalyst showed significant photocatalytic activity toward various organic pollutants [21,23,24,25,26,27]. Most studied molecules are branched with at least one aromatic ring and nitrogen and oxygen atoms in their structure. Degradation pathways and degradation rate is determined by the nature of substituents on aromatic rings [25]. For example, Fu et al. noted that surface OH groups play a vital role in increasing the degradation rate of nitrobenzene and methylene blue after introducing Al_2_O_3_ particles. The structural similarity of previously investigated compounds and selected β-blockers and obtained results in those experiments served as a starting point for this research.

In this paper, we investigate the photocatalytic properties of V_2_O_5_ nanopowder for the degradation of three β-blockers (MET, NAD, and PIN) separately and in a mixture. Several methods of characterization, such as X-ray powder diffraction (XRPD), scanning electron microscopy (SEM), energy-dispersive X-ray spectroscopy (EDS), Brunauer–Emmett–Teller (BET) measurements, Fourier transform infrared (FTIR) and Raman spectroscopy, are employed to correlate structural and morphological properties of synthesized nanopowders and their photocatalytic activity under ultraviolet (UV) irradiation. The efficiency of the photodegradation process was compared with the efficiency of direct photolysis (DF). The experimentally obtained results were correlated with the molecular and periodic DFT analysis to explain the influence of molecular structure on the efficiency of the degradation process.

## 2. Results and Discussion

### 2.1. XRPD

Only one crystalline phase Shcherbinaite, V_2_O_5_ (PDF card no. 01-072-0433; orthorhombic P*mnm* space group) could be identified in the sample, Figure 2. Refined unit cell parameter values (a = 11.506 (2), b = 4.3708 (6), c = 3.5627 (4) Å, V = 179.17 (4) Å^3^) are in good agreement with reference values reported in upper mentioned PDF card (a = 11.5100, b = 4.3690, c = 3.5630 Å, V = 179.173 Å^3^). The significant presence of an amorphous component is not evident. Crystallite-sized (1251(35) Å) and lattice strain (0.16(2)%) values point to well crystalline material with important lattice strain.

### 2.2. SEM/EDS

By applying the hydrothermal method, where ammonium metavanadate was used as a precursor in an acidic media, hexagonal-shaped platelets with well-defined morphology were obtained as a major entity (Figure 3). The diameters of the platelets are in the range of 10–65 μm, whereas their thickness may be estimated to be a few microns. Rashed et al. [28] obtained similar information regarding the morphology, where the cetyltrimethylammonium bromide (CTAB) was used as a surfactant. Due to this, the thinner nanoflake particles were formed with an average diameter of ~70 nm for the sample annealed at 500 °C for 2 h. A similar procedure was conducted by Abdullah et al. [29] with the same surfactant: after annealing at 500 °C for 4 h, the nanoflakes with thicknesses close to 65–80 nm were obtained.

The EDS analysis indicates that the main elements of the catalyst are vanadium and oxygen, and no other impurity elements could be detected (Figure 4). The atomic percentage of V and O in Spectrum 1 is estimated as 39.86% and 60.14%, respectively, indicating possible oxygen deficiency on the edge of the platelet. The values obtained in Spectrum 2, at the flat surface of the platelet, estimated as 25.47% for V and 74.53% for O, are close to the values of stoichiometric V_2_O_5_.

### 2.3. Raman Scattering Measurements

The Raman scattering measurements taken on the catalyst sample have confirmed the results of XRPD regarding orthorhombic α-V_2_O_5_ structure with P*mnm* symmetry. This structure is associated with 21 Raman active modes (7*_Ag_* + 3*B*_1*g*_ + 7*B*_2*g*_ + 4*B*_3*g*_) [30], 10 of which are registered in the spectrum shown in Figure 5. These modes are unambiguously assigned to α-V_2_O_5_ structure [30,31]: ~104(*A_g_*), 146(*B*_1*g*_/*B*_3*g*_), 198 (*A_g_*/*B*_2*g*_), 285 (*B*_1*g*_/*B*_3*g*_), 306 (*A_g_*), 406 (*A_g_*), 483 (*A_g_*), 528 (*A_g_*), 701 (*B*_1*g*_/*B*_2*g*_), and 995 cm^−1^ (*A_g_*). A good agreement of the position of the Raman mode at 995 cm^−1^ with corresponding bulk value, together with the absence of ~840 cm^−1^ mode in the spectra of the catalyst, reveals a good crystallinity of the V_2_O_5_ phase [30,31,32]. A broad feature at ~1020 cm^−1^ may be ascribed to surface VO*_x_* species on V_2_O_5_ [33].

### 2.4. UV-Vis

In order to estimate the energy bandgap of V_2_O_5_ powder, Kubelka–Munk functions *F*(*R*) [34] were calculated as *F*(*R*) = (1 − *R*)^2^/2*R*, where *R* is the diffuse reflectance of the V_2_O_5_ sample. Taking into account that *F*(*R*) is proportional to the absorption coefficient α, the Tauc plot [35] was obtained by using the following equation:$${\left(h\nu \;F\left(R\right)\right)}^{1/n}=A\left(h\nu -E_{g}\right)$$
where *h* is Planck’s constant, *ν*—photon’s frequency, *E*_g_—bandgap, *A*—proportional constant, and the *n* factor depends on the nature of the electron transition. They equal 1/2 or 2 for allowed direct and indirect transition band gaps [36]. The band gap values were deduced by extrapolating the linear portion of the curves to the energy axis, as shown in Figure 6. According to this procedure, the bandgap energy *E*_g_ values, assuming the allowed direct and indirect transitions, are estimated as ~2.24 and ~2.11 eV, respectively.

It should be noted that different V_2_O_5_ structures may undergo direct and indirect transitions, and it is not often easy to decide which type of electron inter-band transition is predominant [37]. According to a detailed analysis of available literature data and the film properties, Schneider concluded that the direct allowed transition could be considered the most probable in investigated V_2_O_5_ films [37]. Conversely, Mousavi et al. [38] have suggested the direct bandgap in the V_2_O_5_ nanoparticles based on the best line fitting of Kubelka–Munk functions. Still, in our case, it is impossible to decide the character of the allowed transition, considering the good quality of line fitting (Figure 6).

In this work, the DFT approach was used to perform band structure calculations to complement our experimental observations and to compare band gaps. The model of V_2_O_5_ was generated according to the previously mentioned information regarding the crystal structure of V_2_O_5._ The model of V_2_O_5_ was first subjected to geometrical optimization, followed by band structure calculations, using the PBE-D3 level of theory with Hubbard U value set to 3.5 eV for vanadium 3d states, a value reported to produce excellent results in terms of band gap for this material [39]. The obtained band structure is presented in Figure 7.

As presented in Figure 7, it can be seen that the DFT calculations lead to excellent matching between experimental and computational results. Our DFT calculations indicate that an indirect band gap characterizes the V_2_O_5_. Additionally, the DFT + U approach utilized in this work yielded a band gap value of 2.289 eV, which is in excellent agreement with our experimental observations and reports of other research groups [40,41]. The importance of Hubbard U correction was essential for reaching this level of agreement. However, it is also interesting to note that the absence of Hubbard U correction led to a reasonably decent band gap value equal to 1.880 eV (again, with the indirect band gap type).

### 2.5. Photocatalytic Application

The photocatalytic activity of the V_2_O_5_ catalyst was studied through the photocatalytic degradation of NAD, PIN, and MET and their mixture. The results of the degradation efficiency were compared with the efficiency of DF.

Since light absorption is an important parameter influencing the degradation efficiency, we have also analyzed the excitations within studied pharmaceutical molecules. For these purposes, we have performed TD-DFT calculations with CAM-B3LYP functional and 6-311++G(d,p) basis set, which enabled us to simulate the UV spectra of the compounds mentioned above (Figure 8). The obtained results agree with the investigated compounds’ experimental spectra.

UV spectra presented in Figure 8 indicate one more interesting property of the PIN molecule, in line with the experimentally observed degradation efficiencies. Namely, aside from the excitations at wavelengths around 200 nm, the PIN molecule is also characterized by relatively strong light absorption at 260 nm. The stronger light absorption of the molecule is significant for photocatalytically assisted degradation and, in the case of the present study, may be a crucial factor leading to the superior degradation efficiency of PIN over other molecules (Figure 9).

The poor degradation efficiency of investigated compounds was expected, considering the investigated compounds’ absorption maxima and the UV lamp’s characteristics. Namely, the absorption maximum of the used lamp corresponds to the UV/A-B region of the spectrum. As shown in Figure 9a, MET and NAD expressed 10% and 15% degradation after 120 min. DF of separate solutions offers slight removal of NAD and MET and a two- to three-times higher degradation rate for PIN (30%). The nature of the substituents attached to the aromatic ring and the secondary interaction of released ions have significant roles in the degradation progress and regulate the degradation pathways. The obtained results can be explained based on the structural characteristics of the tested compounds.

The efficiency of DF is usually improved when irradiation is combined with a photocatalyst. However, V_2_O_5_ did not show a positive effect on the removal of NAD and MET from the aqueous solution (Figure 9b). Moreover, the efficiency of the process has decreased in comparison to DF. We can conclude that the turbidity of the solution resulting from the presence of nanomaterials decreased the removal efficiency of the mentioned two beta blockers. However, PIN showed higher efficiency than MET and NAD in the presence and absence of a V_2_O_5_ catalyst. Namely, 64% of PIN was eliminated by photocatalytic degradation within 120 min.

In addition to the fact that it is necessary to overlap at least partially the radiation spectrum and the compounds that DF decomposes, the pH of the solution plays a vital role in the creation of reactive oxygen species and the further interaction of released ions and radicals. The pH value of the aqueous solutions for DF was about 7.5 ± 1.0 (Figure 10). At the specified pH value of the aqueous solution, reactive oxygen species can be created in the water, contributing to the degradation efficiency.

In the presence of V_2_O_5_, the pH values of water suspensions were around 4.0 ± 0.5 (Figure 10). The effect of pH solution on photocatalytic degradation is complex due to the electrostatic interaction between the semiconductor surface, solvent molecules, substrate, and radicals formed during the photodegradation reaction.

V_2_O_5_ is a transition metal oxide of vanadium that has a narrow band gap (2.3 eV) [22,24]. It can capture a significant fraction of the UV spectrum to generate active redox centers. Photoactivation happens with wavelengths less than 443 nm [42]. Catalyst also accelerated the degradation of PIN two times compared to the DF study within 120 min (64% and 30%, respectively). However, the catalyst did not have expected behavior towards MET and NAD, wherein the degradation rate was almost the same for these two β-blockers (7% and 5%, respectively). Differences between the efficiency of PIN removal and removal of MET and NAD result from differences in these compounds’ structural and electronic properties. With the addition of V_2_O_5,_ the pH value of NAD and PIN solutions decreased from 8.0 to around 3.5. There was no significant change in pH value for the MET solution after V_2_O_5_ addition, and the pH value was about 4.4 during 120 min of photocatalysis.

When the pH is ≥7 ^•^OH radicals are the main reactive species, while at lower pH values, the reactive species are h^+^ [43,44]. Based on pH value, it can be determined through which reactive species the process takes place in the presence of V_2_O_5_. Since the pH value is less than 7 for all three β-blockers, the reactive species are h^+^. However, because the efficiency for MET and NAD is low in the presence of V_2_O_5,_ it implies that degradation did not occur via h^+^. In contrast, the degradation of PIN takes place via h^+^. To better understand the interaction between MET and the ^•^OH radicals, Armaković et al. [45] have conducted a DFT computational analysis. The interaction of ^•^OH radicals with MET formed a new bond. It was concluded that ^•^OH binds to the aromatic ring of MET. Moreover, the structural properties of MET indicated the highest interaction of MET and ^•^OH, compared to other radicals and h^+^. These data are essential, considering that MET and NAD are similar β-blockers, concluding that NAD will also interact with ^•^OH radicals. Data obtained in this work also supported similar behavior of MET and NAD in reaction with V_2_O_5_. Armaković et al. [46] have determined the Fukui *f*_0_ function of the PIN molecule, which provides information about the sensitivity towards radical attacks. Since positive values of Fukui *f*_0_ were mainly located at the PIN rings, they suggested that this molecular site is the most sensitive to radical attacks. However, they emphasized that the h^+^ generated during the photocatalytic process significantly increases degradation efficiency. Jovanoski Kostić et al. [47] have stated that the influence of h^+^ on the photodegradation of PIN is favored in acidic conditions. The obtained pH value (Figure 10) within the photodegradation of PIN using V_2_O_5_ indicated that the mechanism of degradation of PIN occurs mainly via h^+^.

To explain the observed difference in degradation efficiencies, we performed a computational analysis of all molecules in the framework of the DFT approach. Since the degradation efficiency generally depends on the reactivity of molecules and photophysical properties of the catalyst, we decided to compare the reactivity of considered molecules by calculating the average local ionization energy (ALIE), a well-known quantum molecular descriptor. The analysis of the ALIE quantum-molecular descriptor precisely identifies the PIN molecule as the most reactive of all three considered, see Figure 11.

This work visualized the ALIE descriptor as an electron density surface mapped with the ALIE values. ALIE values indicate the energy required to remove an electron from a certain point around the molecule. The lower this energy is, the easier it is to remove an electron from a certain point around the molecule. The results presented in Figure 11 indicate that the PIN is by far the most reactive of all three considered molecules in terms of how easy it is to remove an electron from it. Both MET and NAD have relatively similar lowest values of the ALIE descriptor (the difference is ~4 kcal/mol). On the other side, the PIN has around 14 kcal/mol lower minimal ALIE value than NAD and about 19 kcal/mol lower minimal ALIE value than MET.

This significant difference in ALIE indicates that the PIN molecule is much more sensitive to the influence of positive charge than the other two molecules. This also suggests that the positively charged $h^{+}$ might be responsible for the efficient degradation of the PIN. Since, according to the ALIE descriptor, the other two molecules are not sensitive to positive charge as the PIN is, their degradation efficiency is very low. The distribution of ALIE values on the electron density surface leads to one more important conclusion regarding the PIN’s degradation efficiency. Namely, the lowest ALIE values are delocalized to a much greater extent over the PIN molecule than the MET and NAD. This means that a much higher surface area of the PIN molecule is sensitive toward the positive charge compared to the MET and NAD.

Further, the mixture of NAD, PIN, and MET was subjected to DF and a photocatalytic reaction with V_2_O_5_ to study their degradation when coexisting in environmental waters, as it occurs in nature. As seen in Figure 12a, exposure of the NAD, PIN, and MET mixture to UV did not cause a higher degradation efficiency of these three compounds. The DF of the separate solution of each β-blocker was a more efficient process. The reason may be caused by the additive or synergistic effects of the structures of these three β-blockers and the same amount of UV energy distributed over all three compounds [11]. Moreover, the pH value during decomposition (Figure 13) was constant and slightly basic (around 7.7 ± 0.2), indicating no changes in the aqueous solution.

The lower photocatalytic efficiency of the PIN removal was achieved in a mixture with NAD and MET than in an individual solution. However, a much more interesting result is the increase in the degradation rate of NAD and MET in the presence of PIN (Figure 9b and Figure 12b). One of the possible reasons for the acceleration of the decomposition process of NAD and MET in the mixture with PIN could be the higher concentration of NAD and MET molecules on the surface of the catalyst, which is also an explanation for the lower decomposition efficiency of PIN. Namely, 5 min after the beginning of UV irradiation, a certain amount of each compound was removed, and it continued to slightly increase up to a degradation rate of 13% for NAD, 40% for PIN, and 15% for MET within 120 min.

The mixture solution’s pH was also acidic, but a 1.0 value higher than in the separate pharmaceuticals solutions (around 5.0 ± 0.2). These results also indicate that the presence of all three compounds in water changes starting pH. Moreover, the changing pH during degradation is different, showing the other degradation mechanism and the generation of various intermediates during the degradation process (Figure 10 and Figure 13).

## 3. Materials and Methods

### 3.1. Chemicals and Solutions

The active components of β-blockers, NAD (≥99%, Sigma-Aldrich, Hamburg, Germany), PIN (≥99%, Sigma-Aldrich, Hamburg, Germany), and MET (≥99%, Sigma-Aldrich, Hamburg, Germany) were used as received. All three solutions were made using ultrapure water (UPW, *κ* = 0.055 μS/cm, pH 6.6).

As the catalyst, V_2_O_5_ was used at a concentration of 1.0 mg/cm^3^. Detailed information about the V_2_O_5_ synthesis is given in Section 3.2 and Section 3.3.

Ammonium metavanadate (NH_4_VO_3_, ACS reagent, ≥99.0%, Sigma-Aldrich, Hamburg, Germany), acetic acid (glacial) 100% (CH_3_CO_2_H) anhydrous for analysis (EMSURE ACS, ISO, Reag. Ph Eur, Supelco, Bellefonte, PA, USA), ethanol absolute (CH_3_CH_2_OH, ACS reagent absolute, Supelco) were used as received.

### 3.2. Powder Synthesis

The facile hydrothermal synthesis method was used to prepare V_2_O_5_ powders [14,48]. For that purpose, 0.1 M ammonium metavanadate (NH_4_VO_3_) was dissolved in 80 mL of distilled water, resulting in a pale yellow solution. To adjust the pH of the reaction solution to about 4, the diluted acetic acid (CH_3_COOH/H_2_O = 1:1 *v*/*v*) was added drop-wise. The final orange solution was placed into a Teflon-lined stainless steel autoclave and kept at 180 °C for 24 h and then naturally cooled down to room temperature. The orange product was centrifuged with distilled water and absolute ethanol and dried at 80 °C for 12 h. The as-prepared sample was annealed at 500 °C for 2 h in air.

### 3.3. Characterization

The catalyst was investigated by X-ray powder diffraction (XRPD). Measurements were conducted on a Rigaku Smartlab X-ray diffractometer in θ-θ geometry (the sample in horizontal position) in part focusing on Bragg–Brentano geometry using a D/teX Ultra 250 strip detector in 1D standard mode with a CuKα_1,2_ radiation source (U = 40 kV and I = 30 mA). The XRPD patterns were collected in the 4–90° 2θ range, with a step of 0.01°, and a data collection speed of 5.1 °/min. The low background single crystal silicon sample holder minimizes the background. Unit cell parameters and average crystallite size and lattice strain values were obtained by PDXL2 integrated X-ray powder diffraction software (Version 2.8.30; Rigaku Corporation, Tokyo, Japan).

The morphology and composition/quality of the catalyst were analyzed on SEM (JEOL JSM–6460LV, with the operating voltage of 20 keV) equipped with an EDS INCAx-sight detector and an “INAx-stream” pulse processor (Oxford Instruments, Abingdon, UK).

The Raman scattering spectra of the catalyst were taken in the backscattering geometry at room temperature in the air using Jobin-Yvon T64000 triple spectrometer (with 1800 grooves mm^−1^ grating) equipped with a confocal microscope and a nitrogen–cooled charge-coupled device detector. The spectra were excited by a 514.5 nm line of Ar^+^/Kr^+^ ion laser with an output power of less than 10 mW.

The UV–vis diffuse reflectance (UV–vis DR) spectrum was recorded in the wavelength range of 200−1400 nm using the Shimadzu UV-2600 spectrophotometer equipped with an integrated sphere. The reflectance spectra were measured relative to a reference sample of BaSO_4_.

### 3.4. Photodegradation Experiments

Photocatalytic degradation experiments were conducted through NAD, PIN, and MET solutions (0.05 mmol/dm^3^) exposed to UV irradiation in the presence of catalyst V_2_O_5_ (1.0 mg/cm^3^). In the case of the DF study, solutions were exposed to UV irradiation without a catalyst. Each solution, separately (in the volume of 20 cm^3^), and then a mixture of all three β-blockers, was placed in a cell made of Pyrex glass (total volume ca. 40 cm^3^, liquid layer thickness 35 mm) with a water circulating jacket. The cell was then placed in an ultrasonic bath, and suspensions were sonicated for 15 min. Afterward, a magnetic stir was added to the cell and placed on a magnetic stirrer in the stream of O_2_, thermostated at 25 ± 0.5 °C. The cell has a plain window on which the UV light beam was focused. The UV radiation source was a 125 W high-pressure mercury lamp (emission bands at 290, 293, 296, 304, 314, 335, and 366 nm, with maximum emission at 366 nm and intensity of 2.6 × 10^−^3 W/cm^2^ in the visible region and 1.4 × 10^−2^ W/cm^2^ in the UV region).

In order to investigate the adsorption of selected β-blockers on the surface of the material, a solution of the corresponding compound was analyzed in the presence of V_2_O_5_ under identical conditions as photocatalytic degradation but without irradiation of the solution. The results showed negligible adsorption (less than 1% in 120 min in all three cases).

### 3.5. Analytical Procedure

For each solution (NAD, PIN, and MET) the sample was collected before cell exposure to the UV irradiation, filtered through a Millipore membrane filter (Milex-GV, 0.22 µm), and placed in a chromatographic vial. Then, the samples were collected and filtered after 5, 10, 30, 60, and 120 min from the beginning of exposure to the UV irradiation. At each sample point, the pH value was measured.

In order to study the degradation kinetics, 20 µL of the filtrate was analyzed by Shimadzu UFLC-PDA (Shimadzu Scientific Instruments, Columbia, Maryland, USA) Eclipse XDB-C18 column, 1550 mm × 4.6 mm i.d., particle size 5 µm, 30 °C). The UV/Vis PDA detector was set at wavelengths of maximum absorption of each β-blocker: 210 nm (for NAD), 217 nm (for PIN), and 223 nm (for MET). To achieve better peak separation, gradient elution was used (flow rate 0.7 mL min^−1^, ACN, and water mixture—15% ACN at the beginning increased to 25% ACN at 6 min, and after that, it was constant for the next 2 min; post time 1 min).

### 3.6. Computational Details

All molecular DFT calculations were performed using the B3LYP density functional [49,50,51,52] in combination with a 6-31G(d,p) basis set [53,54,55]. All molecules were subjected to frequency calculations to check that geometrical optimizations identified the true ground states. Frequency calculations yielded only positive values. During geometrical optimizations, frequency, and property calculations, solvent effects (water) were considered in the Poisson–Boltzmann solver framework. Molecular DFT calculations were performed with the Jaguar program [56,57,58,59], as implemented in the Schrödinger Materials Science Suite 2022-2. TD-DFT analyses were performed using the long-range corrected version of B3LYP, namely the CAM-B3LYP density functional [60], together with the 6-311++G(d,p) basis set [61,62]. TD-DFT calculations were performed with ORCA 5.0.3. molecular modeling package developed by Prof. Frank Neese and coworkers [63,64,65,66,67,68,69,70]. Input files for TD-DFT analyses were prepared with the online ORCA input generator of the atomistica.online web application [71], available at https://atomistica.online.

All periodic DFT calculations were performed using the PBE density functional [72], including the empirically derived correction (D3 variant) developed by Prof. Stefan Grimme and coworkers [73,74,75,76,77]. Additionally, we included the Hubbard U correction for vanadium atoms to consider the underestimation of the band gap by the DFT approach [78]. In this case, we set the U value for vanadium to be 3.5 eV [39]. All periodic DFT calculations were performed with the Quantum Espresso program [79,80,81,82], as implemented in the Schrödinger Materials Science Suite 2022-2.

## 4. Conclusions

For the preparation of V_2_O_5_, facile hydrothermal synthesis was used. X-ray diffraction identifies only one crystalline phase in the sample, which is proven by the fact that the crystallite size (1251(35) Å) and lattice strain (0.16(2)%) values point to well-crystallized material with a significant lattice strain. SEM micrographs show that hexagonal-shaped platelets with well-defined morphology were obtained as substantial entities with diameters in the range of 10–65 µm and thicknesses of around a few microns, following previously conducted similar procedures available in the literature. EDS spectra indicate that the main elements of the catalyst are vanadium and oxygen with no other impurities, and the estimated contribution of each component is close to stoichiometric V_2_O_5_. The Raman scattering measurements taken on the catalyst confirmed the result of XRPD.

The efficiency of DF of the individual solutions was poor, except for PIN (10% for MET, 15% for NAD, and 30% for PIN). Adding a catalyst in the water solution of MET and NAD did not improve the degradation. However, the degradation efficiency of PIN increased up to 64% within 120 min. This study considered various perspectives to determine why PIN showed the highest efficiency. The fact that PIN absorbs light at a wavelength of around 200 nm and has relatively strong light absorption at 260 nm may indicate its superior degradation efficiency over the other two molecules. Calculations of ALIE indicate that the PIN is by far the most reactive of all three considered molecules in terms of how easy it is to remove an electron from it. MET and NAD have similar minimal ALIE values, and PIN has around 14 kcal/mol lower minimal ALIE value than NAD and about 19 kcal/mol more down minimal ALIE value than MET. Moreover, the pH value indicated different intermediates of photocatalysis. Namely, following the fact that PIN degradation was at a pH value below 4, it can be concluded that PIN degradation occurs via h^+^.

A photocatalytic study of a mixture of NAD, MET, and PIN, as they occur in nature, shows that the degradation processes for MET and NAD were accelerated compared to individual solutions of these two compounds. This can be explained by a higher concentration of NAD and MET molecules on the surface of the catalyst, which also explains the lower decomposition efficiency of PIN. Conditions in the mixture were acidic, with a pH value higher than in individual solutions. pH values were changing differently, indicating the other degradation mechanism and generation of various intermediates during the degradation process.

This and other research demonstrated that V_2_O_5_ has a significant potential for practical applications to eliminate organic pollutants from the most precious natural resource—water. However, an extensive literature survey performed for this research has indicated several more essential facts regarding the potential of V_2_O_5_ for practical applications. One of the most important facts concerns bandgap engineering in the case of the V_2_O_5_. Namely, it was demonstrated that the bandgap value can be efficiently tuned, either by manipulating the structure of V_2_O_5_ or by applying an external stimulus. It has been reported that the band gap of V_2_O_5_ can either be decreased or increased, depending on the selected technique. The ability to finely tune the band gap of V_2_O_5_ is essential for its practical application as a photocatalyst. Finally, the readily available synthesis of this material is one more factor determining the bright future of the material.

## Figures and Tables

**Figure 1 molecules-28-00655-f001:**
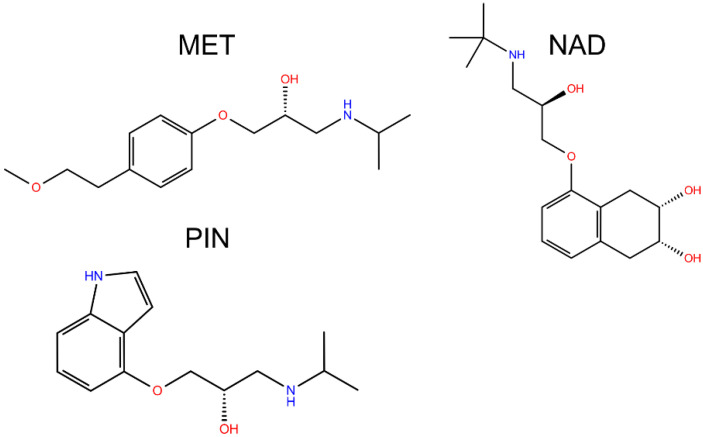
Structural formulas of studied beta blockers.

**Figure 2 molecules-28-00655-f002:**
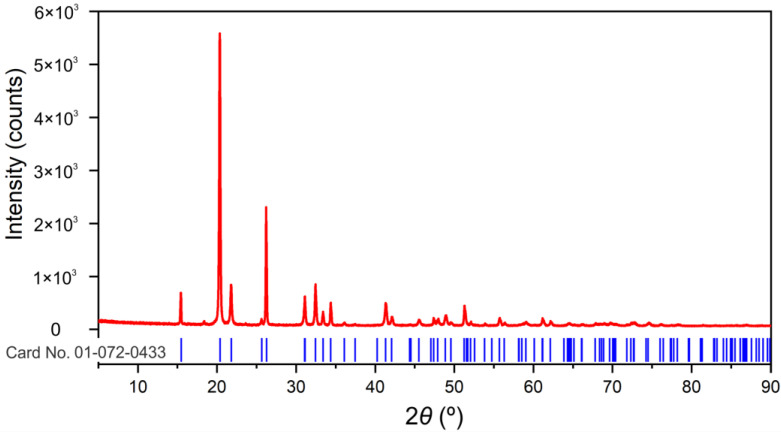
Experimental XRPD pattern in red. Peak positions for Shcherbinaite, V_2_O_5_ (card number 01-072-0433; ICDD (PDF-2 Release 2016 RDB)) in blue are shown below the experimental pattern.

**Figure 3 molecules-28-00655-f003:**
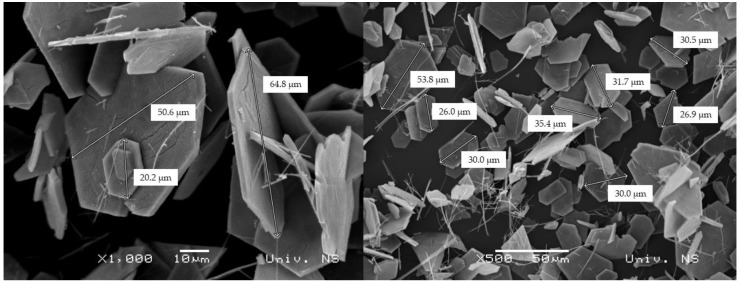
SEM micrographs of V_2_O_5_ powder with different magnifications.

**Figure 4 molecules-28-00655-f004:**
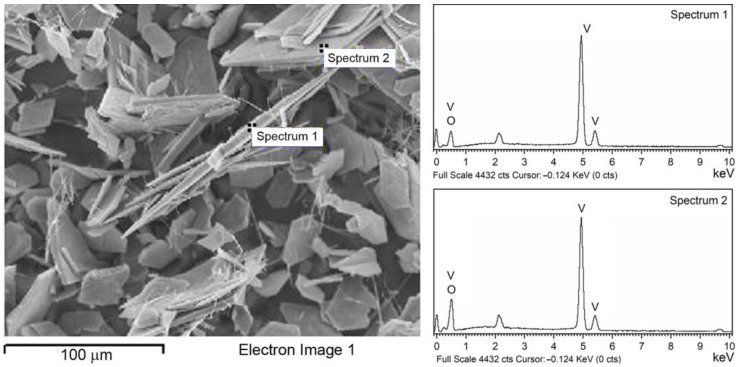
SEM image (**left**) with corresponding EDS spectra (**right**) of V_2_O_5_ catalyst.

**Figure 5 molecules-28-00655-f005:**
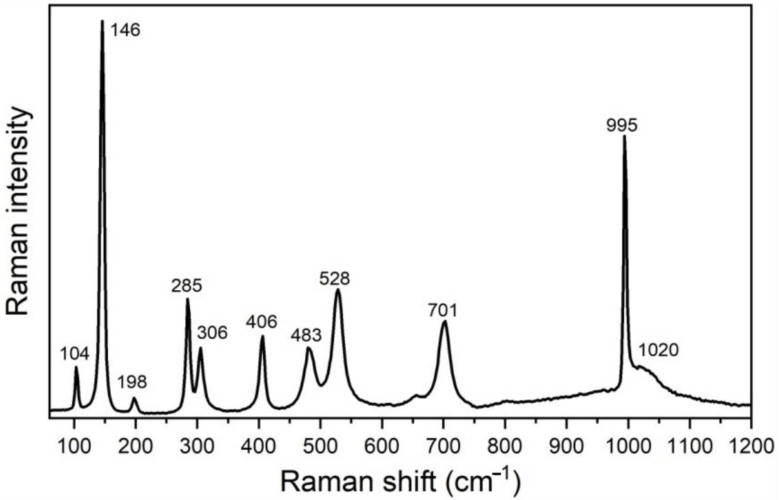
Experimental Raman spectrum of V_2_O_5_ catalyst (after a baseline correction).

**Figure 6 molecules-28-00655-f006:**
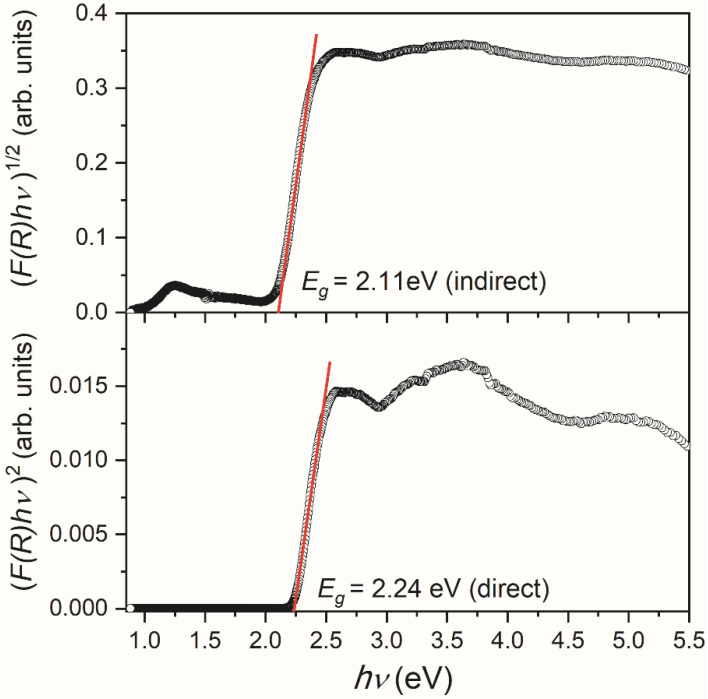
Transformed Kubelka–Munk functions (*F*(*R*) *hν*)^2^ and (*F*(*R*) *hν*)^1/2^ assuming direct and indirect energy bandgap in the V_2_O_5_ sample, respectively.

**Figure 7 molecules-28-00655-f007:**
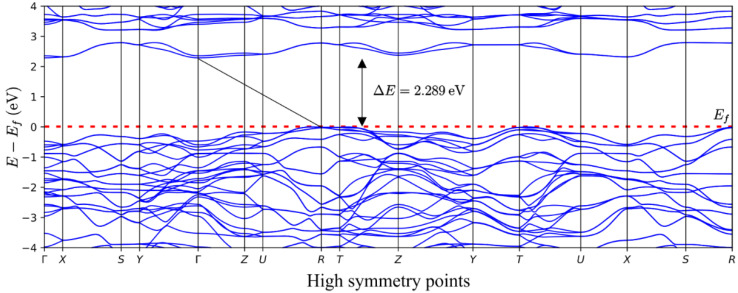
Band structure of V_2_O_5._ The red dotted line denotes the Fermi level. The black line connects the top of the valence and the bottom of the conduction zones.

**Figure 8 molecules-28-00655-f008:**
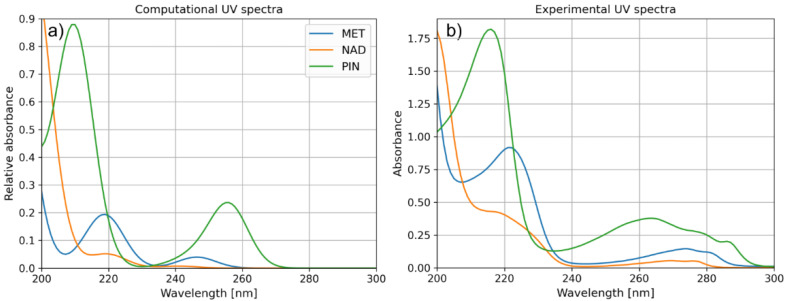
(**a**) Computational and (**b**) experimental UV spectra of MET, NAD, and PIN.

**Figure 9 molecules-28-00655-f009:**
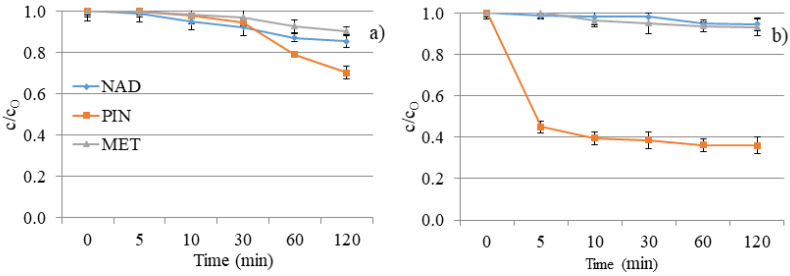
NAD, PIN, and MET degradation kinetics under UV irradiation: (**a**) without V_2_O_5_ and (**b**) with V_2_O_5_ catalyst.

**Figure 10 molecules-28-00655-f010:**
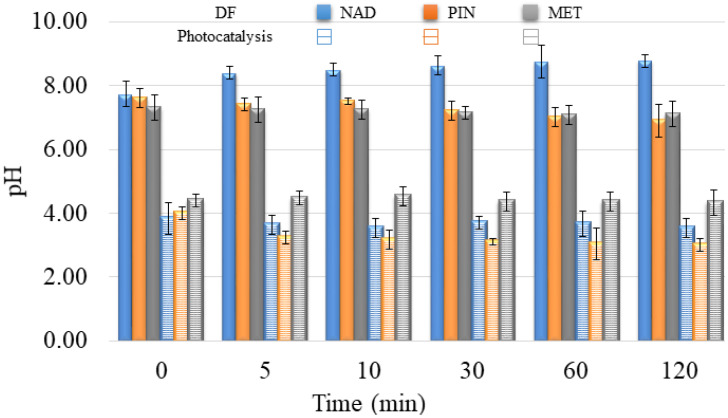
pH value during degradation of selected pharmaceuticals under UV irradiation without V_2_O_5_ (DF) and with V_2_O_5_ catalyst (Photocatalysis).

**Figure 11 molecules-28-00655-f011:**
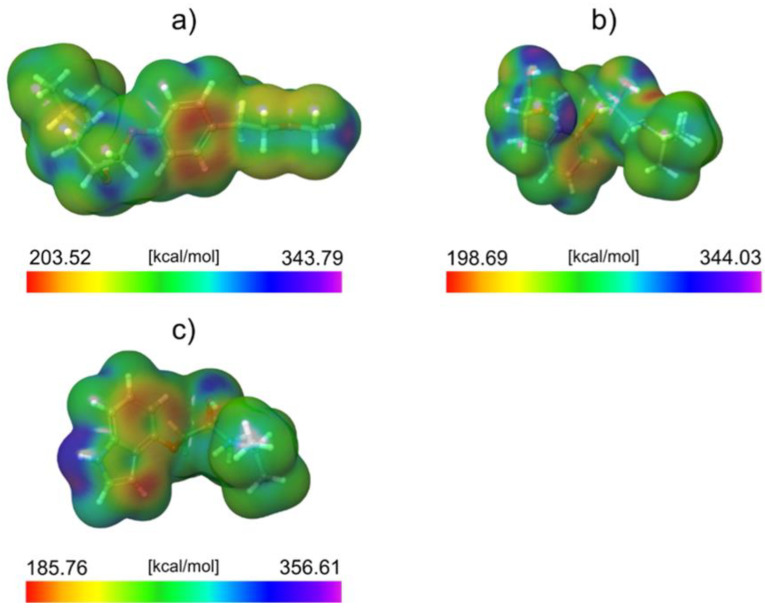
ALIE surfaces of (**a**) MET, (**b**) NAD, and (**c**) PIN.

**Figure 12 molecules-28-00655-f012:**
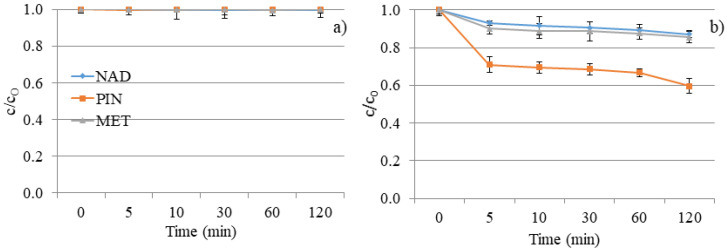
NAD, PIN, and MET mixture degradation kinetics under UV irradiation: (**a**) without V_2_O_5_ and (**b**) with V_2_O_5_ catalyst.

**Figure 13 molecules-28-00655-f013:**
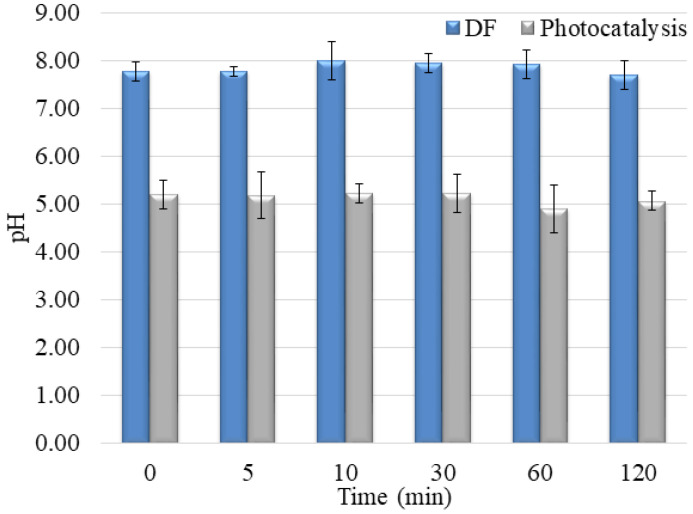
pH values during NAD, PIN, and MET mixture degradation under UV irradiation without/with V_2_O_5_ catalyst.

**Table 1 molecules-28-00655-t001:** Some previous research on the photocatalytic activity of V_2_O_5_ toward organic pollutants.

Synthesis Method	Compounds	Irradiation	Efficiency	Reference
Hydrothermal method V_2_O_5_ pure vanadium pentoxide nanoparticles	Methyl orange (MO) Congo red (CR)	Visible light	After 180 min degradation of MO was 82% and CR was 99.61%	[21]
Coprecipitation—calcination V_2_O_5_/Al_2_O_3_ composite photocatalyst The highest activities were obtained for the sample with the V/Al ratio of 1:1	Reduction of Cr(VI), nitrobenzene (NB), and degradation of methylene blue (MB)	UV	After irradiation for 20, 300, and 20 min, the highest removal (or conversion) efficiencies for Cr(VI), NB, and MB over the optimal sample were found to be 79%, 67%, and 31%, respectively	[23]
Ultrasound-assisted V_2_O_5_ nanoparticles	Rose Bengal dye (RB)	Solar light	After 150 min degradation of RB was around 99%	[24]
Chemical precipitation from ammonium metavanadate using Triton X-100 as surfactant V_2_O_5_ powder	Phenol and derivatives	Natural sunlight	No contribution to the photocatalytic process	[25]
Growing radially on PET fibers V_2_O_5_ nanoflakes	Rhodamine B (RhB)	Visible light	After 60 min degradation of RhB was around 50%	[26]
Simple thermal decomposition method V_2_O_5_/ZnO nanocomposites	MB	Visible light	After 120 min degradation of MB was around 97%	[27]

## Data Availability

Not applicable.

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
