# Peer review of "Photocatalytic Activity of the V2O5 Catalyst toward Selected Pharmaceuticals and Their Mixture: Influence of the Molecular Structure on the Efficiency of the Process"

_molecules, 2023, doi:10.3390/molecules28020655_

Round 1
Reviewer 1 Report
In this manuscript, the authors reported that the photocatalytic activity of V2O5 through the degradation of nadolol (NAD), pindolol (PIN), metoprolol (MET), and their mixture under ultraviolet (UV) irradiation in water. The authors claimed that the photocatalytic degradation of MET and NAD was rather insignificant (<10%). However, in the PIN case, the degradation was significant (64%). In the mixture, the photodegradation efficiency of MET and NAD slightly increased (15% and 13%). On the other side, it reduced for the PIN to the still satisfactory value of 40%. Computational analysis in the framework of density functional theory was used to complement our experimental findings. In overall, this manuscript is interesting but in order to consider publication, this work should be revised. The following comments should be addressed for the improvement of their manuscript.
Comment 1: The overall study aims for this potential photocatalytic activity of V2O5 through the degradation of nadolol (NAD), pindolol (PIN), metoprolol (MET), and their mixture under ultraviolet (UV) irradiation in water need to be further clarified in detail as compared to current conventional system for degradation of nadolol (NAD), pindolol (PIN), metoprolol (MET), and their mixture in our wastewater stream.
Comment 2: The various recent reports and their research findings on the “photocatalytic activity of V2O5 on different persistent organic pollutants” using various potential techniques should be summarized into a table form and discussed for better understanding in term of benchmarking points with your research findings.
Comment 3: The detailed HRETM and XPS analysis can be included to further explain their lattice fridge, chemical state and electronic state of the elements that exist within V2O5 samples.
Comment 4: In addition, the authors should conduct the PL / EIS spectroscopy analysis to provide information such as charge carrier trapping, immigration, and transfer within the V2O5 samples.
Comment 5: The band gap energy and responsive wavelength/illumination of V2O5 samples need to be determined. The particle size, surface area and distribution of V2O5 samples need to be determined too.
Comment 6: The details studies on the standard deviation error bars in data collection in photocatalytic application studies such as NAD, PIN, and MET degradation kinetics, as well as pH value changes need to be provided for better understanding.
Comment 7: The future direction and commercialization perspectives for this potential photocatalysis system with V2O5 samples would adapt to persistent organic pollutants degradation can be further discussed in detail in the conclusion section as compared to current conventional wastewater system.
Comment 8: The carefully English correction is necessary for the whole manuscript. Please check and revise accordingly.
Author Response
Journal Molecules (ISSN 1420-3049)
Manuscript ID molecules-2118619
Title: Photocatalytic activity of V2O5 catalyst toward selected pharmaceuticals and their mixture: Influence of molecular structure on the efficiency of the process
Special Issue Modeling Adsorption Properties of Molecular and Nanostructured Systems for Environmental Applications
The authors thank the reviewer for their constructive comments and recommendations. We have taken the comments on board to improve and clarify the Manuscript. Please find below a detailed point-by-point response to all comments.
The reviewers’ comments are typed in black. The responses to reviewers are tagged blue. The changes in the text of the Manuscript are typed in red. The line numbers in responses to reviewers refer to the corrected text.
Response Reviewer 1:
In this manuscript, the authors reported that the photocatalytic activity of V2O5 through the degradation of nadolol (NAD), pindolol (PIN), metoprolol (MET), and their mixture under ultraviolet (UV) irradiation in water. The authors claimed that the photocatalytic degradation of MET and NAD was rather insignificant (<10%). However, in the PIN case, the degradation was significant (64%). In the mixture, the photodegradation efficiency of MET and NAD slightly increased (15% and 13%). On the other side, it reduced for the PIN to the still satisfactory value of 40%. Computational analysis in the framework of density functional theory was used to complement our experimental findings. In overall, this manuscript is interesting but in order to consider publication, this work should be revised. The following comments should be addressed for the improvement of their manuscript.
We thank the reviewer for the suggestions and the opportunity to improve our manuscript.
Comment 1: The overall study aims for this potential photocatalytic activity of V2O5 through the degradation of nadolol (NAD), pindolol (PIN), metoprolol (MET), and their mixture under ultraviolet (UV) irradiation in water need to be further clarified in detail as compared to current conventional system for degradation of nadolol (NAD), pindolol (PIN), metoprolol (MET), and their mixture in our wastewater stream.
Thank you very much for the suggestion. We improved the quality of our manuscript and compared results with results in the literature. Also, we added adequate references:
[9] L. Bijlsma, E. Pitarch, E. Fonseca, M. Ibanez, A.M. Botero, J. Claros, L. Pastor, F. Hernandez, Investigation of pharmaceuticals in a conventional wastewater treatment plant: Removal efficiency, seasonal variation and impact of a nearby hospital, J. Environ. Chem. Eng. 9 (2021) 105548.
[10] Y. Ye, Y. Feng, H. Bruning, D. Yntema, H. Rijnaarts, Photocatalytic degradation of metoprolol by TiO2 nanotube arrays and UV-LED: Effects of catalyst properties, operational parameters, commonly present water constituents, and photo-induced reactive species, Appl. Catal. B Environ. 220 (2018) 171–181.
[11] A. Stankiewicz, J. Giebułtowicz, U. Stankiewicz, P. Wroczyński, G. Nałęcz-Jawecki, Determination of selected cardiovascular active compounds in environmental aquatic samples–methods and results, a review of global publications from the last 10 years, Chemosphere. 138 (2015) 642–656.
[12] J. Radjenović, M. Petrović, D. Barceló, Fate and distribution of pharmaceuticals in wastewater and sewage sludge of the conventional activated sludge (CAS) and advanced membrane bioreactor (MBR) treatment, Water Res. 43 (2009) 831–841.
[13] H. Liu, M. Chen, D. Wei, Y. Ma, F. Wang, Q. Zhang, J. Shi, H. Zhang, J. Peng, G. Liu, Smart removal of dye pollutants via dark adsorption and light desorption at recyclable Bi2O2CO3 nanosheets interface, ACS Appl. Mater. Interfaces. 12 (2020) 20490–20499.
[14] O.P. Kumar, K. Shahzad, M.A. Nazir, N. Farooq, M. Malik, S.S. Ahmad Shah, A. ur Rehman, Photo-Fenton activated C3N4x/AgOy@Co1-xBi0.1-yO7 dual s-scheme heterojunction towards degradation of organic pollutants, Opt. Mater. (Amst). 126 (2022) 112199.
[15] M. Jamshaid, M.A. Nazir, T. Najam, S.S.A. Shah, H.M. Khan, A. ur Rehman, Facile synthesis of Yb3+-Zn2+ substituted M type hexaferrites: Structural, electric and photocatalytic properties under visible light for methylene blue removal, Chem. Phys. Lett. 805 (2022) 139939. https://doi.org/10.1016/j.cplett.2022.139939.
[16] A.O. Oluwole, E.O. Omotola, O.S. Olatunji, Pharmaceuticals and personal care products in water and wastewater: a review of treatment processes and use of photocatalyst immobilized on functionalized carbon in AOP degradation, BMC Chem. 14 (2020) 1–29.
[18] K. Shahzad, M.I. Khan, A. Shanableh, N. Elboughdiri, S. Jabeen, M.A. Nazir, N. Farooq, H. Ali, A. Abdelfattah, A.U. Rehman, Silver supported-Ag@PMOS onto thumb structured porous organosilica materials with efficient hetero-junction active sites for photo-degradation of methyl orange dye, Inorg. Nano-Metal Chem. 52 (2022) 407–416. https://doi.org/10.1080/24701556.2021.1980021.
[19] R. Suresh, K. Giribabu, R. Manigandan, S. Munusamy, S.P. Kumar, S. Muthamizh, A. Stephen, V. Narayanan, Doping of Co into V2O5 nanoparticles enhances photodegradation of methylene blue, J. Alloys Compd. 598 (2014) 151–160.
[20] S.K. Jayaraj, V. Sadishkumar, T. Arun, P. Thangadurai, Enhanced photocatalytic activity of V2O5 nanorods for the photodegradation of organic dyes: a detailed understanding of the mechanism and their antibacterial activity, Mater. Sci. Semicond. Process. 85 (2018) 122–133.
[21] M.M. Sajid, N.A. Shad, Y. Javed, S.B. Khan, Z. Zhang, N. Amin, H. Zhai, Preparation and characterization of Vanadium pentoxide (V2O5) for photocatalytic degradation of monoazo and diazo dyes, Surf. Interfaces. 19 (2020) 100502.
[22] J. Sun, X. Li, Q. Zhao, J. Ke, D. Zhang, Novel V2O5/BiVO4/TiO2 nanocomposites with high visible-light-induced photocatalytic activity for the degradation of toluene, J. Phys. Chem. C. 118 (2014) 10113–10121.
[23] X. Fu, W. Tang, L. Ji, S. Chen, V2O5/Al2O3 composite photocatalyst: preparation, characterization, and the role of Al2O3, Chem. Eng. J. 180 (2012) 170–177.
[24] K. Karthik, M.P. Nikolova, A. Phuruangrat, S. Pushpa, V. Revathi, M. Subbulakshmi, Ultrasound-assisted synthesis of V2O5 nanoparticles for photocatalytic and antibacterial studies, Mater. Res. Innov. 24 (2020) 229–234.
[25] M. Aslam, I.M. Ismail, N. Salah, S. Chandrasekaran, M.T. Qamar, A. Hameed, Evaluation of sunlight induced structural changes and their effect on the photocatalytic activity of V2O5 for the degradation of phenols, J. Hazard. Mater. 286 (2015) 127–135.
[26] Y.-L. Chan, S.-Y. Pung, S. Sreekantan, Synthesis of V2O5 nanoflakes on PET fiber as visible-light-driven photocatalysts for degradation of RhB dye, J. Catal. 2014 (2014).
[27] R. Saravanan, V. Gupta, E. Mosquera, Fjj. Gracia, Preparation and characterization of V2O5/ZnO nanocomposite system for photocatalytic application, J. Mol. Liq. 198 (2014) 409–412.
Comment 2: The various recent reports and their research findings on the “photocatalytic activity of V2O5 on different persistent organic pollutants” using various potential techniques should be summarized into a table form and discussed for better understanding in term of benchmarking points with your research findings.
We have added Table 1 with literature data and new references to the reference list:
[21] M.M. Sajid, N.A. Shad, Y. Javed, S.B. Khan, Z. Zhang, N. Amin, H. Zhai, Preparation and characterization of Vanadium pentoxide (V2O5) for photocatalytic degradation of monoazo and diazo dyes, Surf. Interfaces. 19 (2020) 100502.
[22] J. Sun, X. Li, Q. Zhao, J. Ke, D. Zhang, Novel V2O5/BiVO4/TiO2 nanocomposites with high visible-light-induced photocatalytic activity for the degradation of toluene, J. Phys. Chem. C. 118 (2014) 10113–10121.
[23] X. Fu, W. Tang, L. Ji, S. Chen, V2O5/Al2O3 composite photocatalyst: preparation, characterization, and the role of Al2O3, Chem. Eng. J. 180 (2012) 170–177.
[24] K. Karthik, M.P. Nikolova, A. Phuruangrat, S. Pushpa, V. Revathi, M. Subbulakshmi, Ultrasound-assisted synthesis of V2O5 nanoparticles for photocatalytic and antibacterial studies, Mater. Res. Innov. 24 (2020) 229–234.
[25] M. Aslam, I.M. Ismail, N. Salah, S. Chandrasekaran, M.T. Qamar, A. Hameed, Evaluation of sunlight induced structural changes and their effect on the photocatalytic activity of V2O5 for the degradation of phenols, J. Hazard. Mater. 286 (2015) 127–135.
[26] Y.-L. Chan, S.-Y. Pung, S. Sreekantan, Synthesis of V2O5 nanoflakes on PET fiber as visible-light-driven photocatalysts for degradation of RhB dye, J. Catal. 2014 (2014).
[27] R. Saravanan, V. Gupta, E. Mosquera, Fjj. Gracia, Preparation and characterization of V2O5/ZnO nanocomposite system for photocatalytic application, J. Mol. Liq. 198 (2014) 409–412.
Comment 3 and 4: The detailed HRETM and XPS analysis can be included to further explain their lattice fridge, chemical state and electronic state of the elements that exist within V2O5 samples. In addition, the authors should conduct the PL / EIS spectroscopy analysis to provide information such as charge carrier trapping, immigration, and transfer within the V2O5 samples.
We agree that additional measurements could be useful, but unfortunately, mentioned experimental methods are not available to us.
Comment 5: The band gap energy and responsive wavelength/illumination of V2O5 samples need to be determined. The particle size, surface area and distribution of V2O5 samples need to be determined too.
The bandgap energy has been estimated by UV-vis measurements and the results are incorporated in the article text - a Figure 5 with corresponding text is inserted as separate paragraph in 2.1.
Regarding particle size and shape, from Figure 2 (SEM/EDS) the diameters hexagonal-shaped platelets were obtained as a major entity estimated in the range of 10 – 65 μm, with the thicknes thickness of a few microns. The thickness of thinner nanoflakes has been estimated as close to 70 nm. Also, from XRPD measurements, an average crystallite size has been estimated as ~ 125 nm.
Comment 6: The details studies on the standard deviation error bars in data collection in photocatalytic application studies such as NAD, PIN, and MET degradation kinetics, as well as pH value changes need to be provided for better understanding.
Thank you very much for noticing this. As reviewers requested, we added standard deviation error bars in degradation kinetics of photodegradation of NAD, PIN, and MET, as well as graphs of pH value changes.
Comment 7: The future direction and commercialization perspectives for this potential photocatalysis system with V2O5 samples would adapt to persistent organic pollutants degradation can be further discussed in detail in the conclusion section as compared to current conventional wastewater system.
Thank you very much for mentioning this. For these purposes, we have added one more paragraph in the concluding section of the revised version of our manuscript.
Comment 8: The carefully English correction is necessary for the whole manuscript. Please check and revise accordingly.
We accepted the reviewer's suggestion, and with the help of a colleague who was skillful in writing scientific texts in English, we improved the quality of the English language in our manuscript.

Reviewer 2 Report
Manuscript ID: molecules-2118619
Title: Photocatalytic activity of V2O5 catalyst toward selected pharmaceuticals and their mixture: Influence of molecular structure on the efficiency of the process
Comments.
Sanja J. Armaković and co-authors choose interesting topic of degradation of pharmaceutical pollutants (β-blockers) with V2O5. Although the topic is interesting, but some important aspects were not performed also the manuscript setting and formatting is not according to standard. Following comments should be addressed before possible consideration for publication in worthy Journal of Molecules. I believe it will not take a long for the authors to work on this revision. My comments are,
1. In abstract, remove less important sentences in stating lines and add quantitative data
2. In introduction line 41 correct word waters as water.
3. Introduction is too short. Authors have not discuss different treatment techniques. Also authors hane not clear, why they opt photocatalysis technique for the removal of pharmaceuticals. In introduction more literature should be reviewed and some latest photocatalyts should be discussed here to enhance the novelty of work like, https://doi.org/10.1080/24701556.2021.1980021, Optical Materials 126 (2022) 112199, Chemical Physics Letters 805 (2022) 139939
4. Section 2.1, Powder synthesis should not be under results and discussion section. Move it to materials and methods section
5. Section 2.2, Characterization methods should be as Characterization.
6. Figure 1 is not correct, set values along y axis, also remove border line along x axis or separate spectral line from border line
7. Fig 3 increase the size of EDS spectra to be visible clearly
8. Fig 7 is not understand able, efficiency may checked for different beta blockers at different pH, but this figure not showing such information.
9. Original absorbance graphs/UV-vis spectra of degradation of NAD, PIN, and MET should be added
10. Authors should calculate the band gap of V2O5
11. Section 3.1and 3.2 should be merged.
12. Reusability test should be performed
13. Proposed mechanism for the degradation of NAD, PIN, and MET with V2O5 should be discussed in detail.
14. FTIR spectra of V2O5 before and after photocatalysis should be carried out
15. There are so many typo grammatical errors in whole manuscript, should be revised by some native speaker and formatting should be checked.
Author Response
Journal Molecules (ISSN 1420-3049)
Manuscript ID molecules-2118619
Title: Photocatalytic activity of V2O5 catalyst toward selected pharmaceuticals and their mixture: Influence of molecular structure on the efficiency of the process
Special Issue Modeling Adsorption Properties of Molecular and Nanostructured Systems for Environmental Applications
The authors thank the reviewer for their constructive comments and recommendations. We have taken the comments on board to improve and clarify the Manuscript. Please find below a detailed point-by-point response to all comments.
The reviewers’ comments are typed in black. The responses to reviewers are tagged blue. The changes in the text of the Manuscript are typed in red. The line numbers in responses to reviewers refer to the corrected text.
Response Reviewer 2:
Sanja J. Armaković and co-authors choose interesting topic of degradation of pharmaceutical pollutants (β-blockers) with V2O5. Although the topic is interesting, but some important aspects were not performed also the manuscript setting and formatting is not according to standard. Following comments should be addressed before possible consideration for publication in worthy Journal of Molecules. I believe it will not take a long for the authors to work on this revision.
The authors thank the Reviewer very much for this comment and positive opinion of our results.
- In abstract, remove less important sentences in stating lines and add quantitative data.
Thank you very much for this point. We rewrote the Abstract.
- In introduction line 41 correct word waters as water.
We thank the Reviewer for spotting this error.
- Introduction is too short. Authors have not discuss different treatment techniques. Also authors hane not clear, why they opt photocatalysis technique for the removal of pharmaceuticals. In introduction more literature should be reviewed and some latest photocatalyts should be discussed here to enhance the novelty of work like, https://doi.org/10.1080/24701556.2021.1980021, Optical Materials 126 (2022) 112199, Chemical Physics Letters 805 (2022) 139939
Thank you very much for noticing this. As reviewers requested, in the Introduction section, we have discussed different treatment techniques.
We also added new references to the reference list.
[9] L. Bijlsma, E. Pitarch, E. Fonseca, M. Ibanez, A.M. Botero, J. Claros, L. Pastor, F. Hernandez, Investigation of pharmaceuticals in a conventional wastewater treatment plant: Removal efficiency, seasonal variation and impact of a nearby hospital, J. Environ. Chem. Eng. 9 (2021) 105548.
[10] Y. Ye, Y. Feng, H. Bruning, D. Yntema, H. Rijnaarts, Photocatalytic degradation of metoprolol by TiO2 nanotube arrays and UV-LED: Effects of catalyst properties, operational parameters, commonly present water constituents, and photo-induced reactive species, Appl. Catal. B Environ. 220 (2018) 171–181.
[11] A. Stankiewicz, J. Giebułtowicz, U. Stankiewicz, P. Wroczyński, G. Nałęcz-Jawecki, Determination of selected cardiovascular active compounds in environmental aquatic samples–methods and results, a review of global publications from the last 10 years, Chemosphere. 138 (2015) 642–656.
[12] J. Radjenović, M. Petrović, D. Barceló, Fate and distribution of pharmaceuticals in wastewater and sewage sludge of the conventional activated sludge (CAS) and advanced membrane bioreactor (MBR) treatment, Water Res. 43 (2009) 831–841.
[13] H. Liu, M. Chen, D. Wei, Y. Ma, F. Wang, Q. Zhang, J. Shi, H. Zhang, J. Peng, G. Liu, Smart removal of dye pollutants via dark adsorption and light desorption at recyclable Bi2O2CO3 nanosheets interface, ACS Appl. Mater. Interfaces. 12 (2020) 20490–20499.
[14] O.P. Kumar, K. Shahzad, M.A. Nazir, N. Farooq, M. Malik, S.S. Ahmad Shah, A. ur Rehman, Photo-Fenton activated C3N4x/AgOy@Co1-xBi0.1-yO7 dual s-scheme heterojunction towards degradation of organic pollutants, Opt. Mater. (Amst). 126 (2022) 112199.
[15] M. Jamshaid, M.A. Nazir, T. Najam, S.S.A. Shah, H.M. Khan, A. ur Rehman, Facile synthesis of Yb3+-Zn2+ substituted M type hexaferrites: Structural, electric and photocatalytic properties under visible light for methylene blue removal, Chem. Phys. Lett. 805 (2022) 139939. https://doi.org/10.1016/j.cplett.2022.139939.
[16] A.O. Oluwole, E.O. Omotola, O.S. Olatunji, Pharmaceuticals and personal care products in water and wastewater: a review of treatment processes and use of photocatalyst immobilized on functionalized carbon in AOP degradation, BMC Chem. 14 (2020) 1–29.
[18] K. Shahzad, M.I. Khan, A. Shanableh, N. Elboughdiri, S. Jabeen, M.A. Nazir, N. Farooq, H. Ali, A. Abdelfattah, A.U. Rehman, Silver supported-Ag@PMOS onto thumb structured porous organosilica materials with efficient hetero-junction active sites for photo-degradation of methyl orange dye, Inorg. Nano-Metal Chem. 52 (2022) 407–416. https://doi.org/10.1080/24701556.2021.1980021.
[19] R. Suresh, K. Giribabu, R. Manigandan, S. Munusamy, S.P. Kumar, S. Muthamizh, A. Stephen, V. Narayanan, Doping of Co into V2O5 nanoparticles enhances photodegradation of methylene blue, J. Alloys Compd. 598 (2014) 151–160.
[20] S.K. Jayaraj, V. Sadishkumar, T. Arun, P. Thangadurai, Enhanced photocatalytic activity of V2O5 nanorods for the photodegradation of organic dyes: a detailed understanding of the mechanism and their antibacterial activity, Mater. Sci. Semicond. Process. 85 (2018) 122–133.
[21] M.M. Sajid, N.A. Shad, Y. Javed, S.B. Khan, Z. Zhang, N. Amin, H. Zhai, Preparation and characterization of Vanadium pentoxide (V2O5) for photocatalytic degradation of monoazo and diazo dyes, Surf. Interfaces. 19 (2020) 100502.
[22] J. Sun, X. Li, Q. Zhao, J. Ke, D. Zhang, Novel V2O5/BiVO4/TiO2 nanocomposites with high visible-light-induced photocatalytic activity for the degradation of toluene, J. Phys. Chem. C. 118 (2014) 10113–10121.
[23] X. Fu, W. Tang, L. Ji, S. Chen, V2O5/Al2O3 composite photocatalyst: preparation, characterization, and the role of Al2O3, Chem. Eng. J. 180 (2012) 170–177.
[24] K. Karthik, M.P. Nikolova, A. Phuruangrat, S. Pushpa, V. Revathi, M. Subbulakshmi, Ultrasound-assisted synthesis of V2O5 nanoparticles for photocatalytic and antibacterial studies, Mater. Res. Innov. 24 (2020) 229–234.
[25] M. Aslam, I.M. Ismail, N. Salah, S. Chandrasekaran, M.T. Qamar, A. Hameed, Evaluation of sunlight induced structural changes and their effect on the photocatalytic activity of V2O5 for the degradation of phenols, J. Hazard. Mater. 286 (2015) 127–135.
[26] Y.-L. Chan, S.-Y. Pung, S. Sreekantan, Synthesis of V2O5 nanoflakes on PET fiber as visible-light-driven photocatalysts for degradation of RhB dye, J. Catal. 2014 (2014).
[27] R. Saravanan, V. Gupta, E. Mosquera, Fjj. Gracia, Preparation and characterization of V2O5/ZnO nanocomposite system for photocatalytic application, J. Mol. Liq. 198 (2014) 409–412.
- Section 2.1, Powder synthesis should not be under results and discussion section. Move it to materials and methods section
We thank the Reviewer for this suggestion. Now, section 2.1 is section 3.2.
- Section 2.2, Characterization methods should be as Characterization.
We corrected it.
6 Figure 1 is not correct, set values along y axis, also remove border line along x axis or separate spectral line from border line
We appreciate noticing, the Figure 1 is improved according to the referee's remarks.
7 Fig 3 increase the size of EDS spectra to be visible clearly.
We thank the Reviewer for this suggestion. The Figure 3 containing EDS data is improved in order to be visible.
8 Fig 7 is not understand able, efficiency may checked for different beta blockers at different pH, but this figure not showing such information.
Thank you very much for this comment. In the revised version of the manuscript, we have made efforts to change Figure 7 so that the information is more straightforward.
9 Original absorbance graphs/UV-vis spectra of degradation of NAD, PIN, and MET should be added
Thank you very much for this comment. In the manuscript's revised version, we added the original absorbance spectra of NAD, PIN, and MET in Figure 5.
10 Authors should calculate the band gap of V2O5
Thank you for suggesting this. The band structure of V2O5 was calculated using the periodic DFT calculations and the computational details section is updated with corresponding details. We obtained excellent agreement between experimentally and computational band gaps.
11 Section 3.1and 3.2 should be merged.
Thank you very much for this suggestion. We merged them.
12 and 14 Reusability test should be performed. FTIR spectra of V2O5 before and after photocatalysis should be carried out.
Unfortunately, we did not investigate the cyclic test, but we understand entirely the rewiever’s point. We will definitively have this in mind for cyclic tests in future studies.
13 Proposed mechanism for the degradation of NAD, PIN, and MET with V2O5 should be discussed in detail.
Thank you very much for bringing this out. We have performed some additional literature surveys and discussed in detail the mechanism of degradation of NAD, PIN, and MET with V2O5.
We also added new references to the reference.
[45] S.J. Armaković, S. Armaković, N.L. Finčur, F. Šibul, D. Vione, J.P. Šetrajčić, B.F. Abramović, Influence of electron acceptors on the kinetics of metoprolol photocatalytic degradation in TiO2 suspension. A combined experimental and theoretical study, RSC Adv. 5 (2015) 54589–54604. https://doi.org/10.1039/c5ra10523d.
[46] S.J. Armaković, S. Armaković, F. Šibul, D.D. Četojević-Simin, A. Tubić, B.F. Abramović, Kinetics, mechanism and toxicity of intermediates of solar light induced photocatalytic degradation of pindolol: Experimental and computational modeling approach, J. Hazard. Mater. (2020) 122490.
[47] A. Jovanoski Kostić, N. Kanas, V. Rajić, A. Sharma, S.S. Bhattacharya, S. Armaković, M.M. Savanović, S.J. Armaković, Evaluation of Photocatalytic Performance of Nano-Sized Sr0.9La0.1TiO3 and Sr0.25Ca0.25Na0.25Pr0.25TiO3 Ceramic Powders for Water Purification, Nanomaterials. 12 (2022). https://doi.org/10.3390/nano12234193.
15 There are so many typo grammatical errors in whole manuscript, should be revised by some native speaker and formatting should be checked.
We accepted the reviewer's suggestion, and with the help of a colleague who was skillful in writing scientific texts in English, we improved the quality of the English language in our manuscript.

Reviewer 3 Report
In this study, the authors have presented their investigations on the photocatalytic activity of V2O5 for the degradation of organic pharmaceutical pollutants, nadolol, pindolol, metoprolol, and their mixture under ultraviolet irradiation in water. Their findings conclude that V2O5 catalyst has no effect on the degradation of nadolol and metoprolol, whereas pindolol shows a significant degradation under the condition provided. It is interesting to see that presence of Pindolol increases the degradation rate of nadolol and metoprolol in the mixture of pollutants.
The overall study is well organized and has been presented in clear and attractive manner. Utilization of computational calculations to explain the observed results is quite impressive. Considering these factors this reviewer would like to recommend publishing this work without further revisions.
Author Response
Journal Molecules (ISSN 1420-3049)
Manuscript ID molecules-2118619
Title: Photocatalytic activity of V2O5 catalyst toward selected pharmaceuticals and their mixture: Influence of molecular structure on the efficiency of the process
Special Issue Modeling Adsorption Properties of Molecular and Nanostructured Systems for Environmental Applications
The authors thank the reviewer for their constructive comments and recommendations. We have taken the comments on board to improve and clarify the Manuscript. Please find below a detailed point-by-point response to all comments.
The reviewers’ comments are typed in black. The responses to reviewers are tagged blue. The changes in the text of the Manuscript are typed in red. The line numbers in responses to reviewers refer to the corrected text.
Response Reviewer 3:
In this study, the authors have presented their investigations on the photocatalytic activity of V2O5 for the degradation of organic pharmaceutical pollutants, nadolol, pindolol, metoprolol, and their mixture under ultraviolet irradiation in water. Their findings conclude that V2O5 catalyst has no effect on the degradation of nadolol and metoprolol, whereas pindolol shows a significant degradation under the condition provided. It is interesting to see that presence of Pindolol increases the degradation rate of nadolol and metoprolol in the mixture of pollutants.
The overall study is well organized and has been presented in clear and attractive manner. Utilization of computational calculations to explain the observed results is quite impressive. Considering these factors this reviewer would like to recommend publishing this work without further revisions.
We thank the reviewer for the time and expertise in reviewing our manuscript. We also appreciate the positive feedback and comments that motivate us to give our best in future research efforts.

Round 2
Reviewer 1 Report
In overall, this manuscript was technically well revised. This revised manuscript meets the criteria of Molecules. Therefore, in my opinion, the revised manuscript can be accepted for publication.
Author Response
Journal Molecules (ISSN 1420-3049)
Manuscript ID molecules-2118619
Title: Photocatalytic activity of V2O5 catalyst toward selected pharmaceuticals and their mixture: Influence of molecular structure on the efficiency of the process
Special Issue Modeling Adsorption Properties of Molecular and Nanostructured Systems for Environmental Applications
Response Reviewer 1:
The authors thank the reviewer for the time and expertise in reviewing our manuscript. We also appreciate the positive feedback and comments that motivate us to give our best in future research efforts.

Reviewer 2 Report
Accept
Author Response
Journal Molecules (ISSN 1420-3049)
Manuscript ID molecules-2118619
Title: Photocatalytic activity of V2O5 catalyst toward selected pharmaceuticals and their mixture: Influence of molecular structure on the efficiency of the process
Special Issue Modeling Adsorption Properties of Molecular and Nanostructured Systems for Environmental Applications
Response Reviewer 2:
The authors thank the reviewer for the time and expertise in reviewing our manuscript. We also appreciate the positive feedback and comments that motivate us to give our best in future research efforts.
